# Evaluating the Impact of Intraoperative MRI in Neuro-Oncology by Scientometric Analysis

**DOI:** 10.3390/life12020175

**Published:** 2022-01-25

**Authors:** Harsh Deora, Gianluca Ferini, Kanwaljeet Garg, M. D. Krishna Narayanan, Giuseppe Emmanuele Umana

**Affiliations:** 1Department of Neurosurgery, National Institute of Mental Health and Neurosciences, Bangalore 560029, India; harshd@nimhans.ac.in; 2REM Radioterapia srl, Viagrande, 95029 Catania, Italy; gianluca.ferini@grupposamed.com; 3Department of Neurosurgery, All-India Institute of Medical Sciences, New Delhi 110029, India; 4Department of Neurosurgery, Yenepoya Medical College, Mangalore 574142, India; mdknarayanan@gmail.com; 5Department of Neurosurgery, Trauma and Gamma-Knife Center, Cannizzaro Hospital, 95126 Catania, Italy; umana.nch@gmail.com

**Keywords:** intraoperative MRI, bibliometric, citations, article impact, neuro-oncology

## Abstract

(1) Objective—Intraoperative Magnetic Resonance Imaging (IOMRI) guided surgery has revolutionized neurosurgery and has especially impacted the field of Neuro-Oncology, with randomized controlled trails demonstrating improved resection, fewer postoperative deficits and enhanced survival rates. Bibliometric analysis allows for analysing chronological trends and measuring the impact and directions of research in a particular field. To the authors’ knowledge, this is the first Bibliometric analysis conducted on IOMRI. (2) Methods—a title specific search of the *Web of Science* database was executed using the keywords ‘intraoperative MRI’, ‘intraoperative magnetic resonance imaging’, and “IOMRI’ on 23rd April 2021. Results—663 articles met the inclusion criteria and were included in the final analysis. In addition, the 100 most cited were analysed as well. Among these 100 articles, 76 were original research papers, while 14 others were review articles. Amongst all the authors, Ganslandt contributed the maximum number of articles, with USA being the largest single source of these articles, followed by Germany. Interestingly, a shift of trends from “Image guided surgery’ and ‘accuracy’ in the early 2000s to ‘extent of resection’, ‘impact’, and ‘survival’ in the later years was noted. (3) Conclusions—IOMRI has now become an integral part of neurosurgery, especially in neuro-oncology. Focus has now shifted from implementation to refinement of technique in the form of functional and oncological outcomes. Therefore, future research in this direction is imperative and will be of more impact that in any other sub-field related to IOMRI.

## 1. Introduction

Neurosurgery has always been driven by the tenets of addressing pathology while maximizing precision, accuracy and, by extension, minimizing morbidity. Neuro-navigation and the availability of image-based guidance while performing surgery has partially addressed this dictum. However, as these are reliant on preoperative images, the accuracy progressively declines during surgery, due to brain shift. From here, the surgeon must rely on his or her own visual cues and judiciously choose a further course of action. The development of the first intraoperative MRI, in 1991, addressed this challenge [1]. It was apparent that intraoperative MRI (IOMRI) images were significantly superior, as they provided accurate information on residual/resected pathology and its relationship with the brain. Although other intraoperative tools, such as ultrasonography (USG) and computed tomography (CT), are available, the image quality and interpretive ability of an MRI image is far superior [1].

A prototype intraoperative MRI unit was first installed in 1994 at the Brigham & Women’s Hospital [1]. Subsequently, the first IOMRI guided stereotactic biopsy was undertaken in June 1995, and in August 1996, the IOMRI was first used in the resection of a brain tumour [1]. In 1997 and in 1999, Black and Jolesz summarized their initial experiences with the development, implementation, and the neurosurgical applications of IOMRI in two landmark papers published in *Neurosurgery.* This set the stage for a more widespread use and acceptance of IOMRI in neurosurgical procedures [2,3]. This has been followed by an extensive use of IOMRI, the world over, with several landmark articles and randomised articles cementing the utility and superiority of this technique.

This article aims to identify the available literature on IOMRI, with a further emphasis on the analyses of the 100 most influential articles on IOMRI by utilizing citation and bibliometric analyses. Citation analysis is a true measure of the impact of articles with several factors influencing the same [4]. Introduced in 1969, bibliometric analysis is superior to peer reviewed eminence-based analysis [5]. It can summarize the chronology and objective citation patterns, along with areas, authors, and journals with the highest impact and the greatest number of publications in a particular topic. There have been a few bibliometric analyses conducted previously in Neurosurgery, and they have proven to be a true reflection of the topic discussed [5].

## 2. Methods

### 2.1. Search Strategy

A title specific search of the *Web of Science* database was executed using the keywords ‘intraoperative MRI’, ‘intraoperative magnetic resonance imaging’, and “IOMRI’ on 23rd April 2021. All the abstracts were screened for the suitable articles. The inclusion criteria were articles relevant to the use of Intraoperative MRI in neurosurgery and published in peer reviewed journals. In addition to the available literature, the 100 most cited articles were selected and reviewed by the authors. All the articles published on Intraoperative MRI and the top 100 cited articles were analysed separately. Bibliometric analysis may sometimes be superior to, or more detailed than, “human-hand search” as it is a computer base comprehensive search and has analysis trends that are not visible in a manual one.

### 2.2. Data

The articles were arranged in descending order according to the number of citations. The parameters assessed were title of the articles, authors, corresponding authors, country of origin, journal of publication, year of publication, citation count, and the journal impact factor.

### 2.3. Analysis

The statistical analysis was performed using R language v 4.0.3 (R Foundation for Statistical Computing, Vienna, Austria) employing the “bibliometrix’ package [6]. The VOSviewer software (Van Eck and Waltman, Leiden University, Leiden, The Netherlands) was also used to plot network and overlay plots [7].

## 3. Results

Among the articles on “Intraoperative MRI” (as per the Web of Science database) published between 1996 and 2020, 663 met the inclusion and exclusion criteria and were included in the final analysis (Table 1). The 100 most cited articles were selected using the “Times cited” feature on Web of Science and analysed separately. These articles ranged in the year they were published, from 1996 to 2016 (Table 1). Interestingly, 76 out of these 100 most cited articles constituted original research papers, while 14 others were review articles (Table 1).

The highest number of articles on IOMRI (overall), published in a single year, were published in 2016 (*n* = 52) (Figure 1A), whereas the highest number of articles, among the 100 most cited articles, was published in 2005 (12%) (Figure 1B) with most of them having been published between 1996 and 2005 (67%). This may, in turn, be due to the number of citations garnered by these articles with the progression of years [8]. More importantly, the maximum number of mean citations per article and mean citations per year, for all the articles, were seen for the article published in 1996, while the articles published in 1997 received the highest number of mean citations per article and mean citations per year. These are markers of the most influential articles produced, independent of the number of citable years (Figure 1C).

A point to note here is that, although Figure 1A shows reduced scientific production after 2017, the reason is that the top 100 cited articles were from 1996 to 2016, and later, citation declined due to recency of their publication. This should not be interpreted as reduced scientific interest in intraoperative MRI studies.

### 3.1. Authors

Among the 663 articles overall, 51 papers were single author papers, while the remaining involved multiple authors. While considering the 100 most cited articles, a similar trend was observed with only four single author papers (Table 1). While 1984 authors contributed to 663 articles, the top 100 cited articles were authored by a total of 350 authors (Table 1). An average of 6.13 co-authors were present for each article in the top 100 cited group, with a collaboration index of 3.61 (Table 1). Most of the authors (88.6%) contributed to either 1 or 2 articles.

Amongst the top 100 cited articles, O Ganslandt contributed to the maximum number of articles (*n* = 24), followed by Nimsky C (*n* = 22) (Figure 2A, Appendix A). Among all the articles published on the topic, Nimsky C has had the maximum impact with a h index of 23 and, consequently, has the highest G and M index (Figure 2B, Table 2). The G-Index is wherein the top G articles have, together, received G citations. The M-Index is the H-index divided by the number of years that a scientist has been active. These indices allow for a weighted analysis of the authors’ contributions. Nimsky C also began publishing early on this topic (1998) and has accrued 2218 citations to-date on IOMRI related scientific publications (Figure 2C, Table 2). O Ganslandt and C Nimsky had the highest number of articles produced on IOMRI in a single year (*n* = 5), both in the year 2005 (Figure 2C). They also had the highest number of citations per year of 41.438. The average citation per article, in the top 100 cited group, was 113.5, which translated to 7.444 citations per document per year (Table 1). A total of 2089 references were used in these 100 articles.

### 3.2. Journals

*Neurosurgery* journal has published the maximum number of articles on IOMRI (*n*= 145) (Figure 3A). It was followed by *Neuro-oncology* (*n*= 51)*, Journal of Neurosurgery*, and *World Neurosurgery* (*n* = 41 each) (Table 3). Among the top 100 cited articles, *Neurosurgery* (29% of articles), again, had the highest number of publications and was followed by the *Journal of Neurosurgery* (10% of articles), *Radiology* (5% of articles), and the *Journal of Magnetic resonance imaging* (4% of articles).

Bradford’s law is a pattern, first described by Samuel C. Bradford in 1934, that estimates the exponentially diminishing results of searching for references in scientific journals [9]. *Neurosurgery* and *Journal of Neurosurgery* were in the zone 1 of Bradford’s law for the top 100 cited articles (Figure 3B). Considering all the articles published on IOMRI, *Neurosurgery, Neuro-oncology*, and *Journal of Neurosurgery* were in the zone 1 of Bradford’s law, and World Neurosurgery was in the zone 2 of Bradford’s law.

Among all the articles on IOMRI, the most cited journal was Neurosurgery, Journal of Neurosurgery, Acta Neurochirurgica, Radiology, American Journal of Neuroradiology (Figure 4A).

*Neurosurgery* had the highest h and g index of 29 and 64, respectively, followed by the *Journal of Neurosurgery* (17,28) and *World Neurosurgery* (12,13,18) (Figure 4B, Table 4). Total citations received by these three journals were 4189, 833, and 377, respectively. Figure 4C shows the year-wise cumulative number of publications on IOMRI in the top 6 journals, which published the maximum number of articles. The curve is steep for *Neurosurgery* in the 1990s and for *World Neurosurgery* after 2010.

As expected, the most frequently used keywords were “resection”, followed by “extent” and “surgery” (Appendix A). This, against the previous trend of “Image guided surgery” and “accuracy”, indicates an increased belief in the mechanism, use of intraoperative MRI, and focus on the potential benefit to the patients.

### 3.3. Countries and Institutions

The most significant contributing countries (in terms of the country of the corresponding author) overall were the United States of America (*n* = 184), Germany (*n* = 124), and China (*n* = 38) (Figure 5A). The orange colour denotes the publication resulting from collaboration of authors from multiple countries. When considering the 100 most cited articles, most of the articles originated from the United States of America (USA) (*n* = 42), closely followed by Germany (*n* = 36), and then the United Kingdom (*n* = 4) (Figure 5B). On a similar note, the United States of America was the country that garnered the most citations per document, with a total of 5053 citations and an average of 120.3 citations for each article (Figure 5C). Interestingly, Netherlands had a higher average per article citation of 215, followed by Israel with 156. The University of Erlangen–Nuremberg was responsible for the highest number of these articles (44 articles), followed by Harvard University (39 articles) and the University of Ulm (32 articles) (Figure 5D).

### 3.4. Most Cited Articles

The most cited article globally was by Zou KH et al., titled “Statistical validation of image segmentation quality based on a spatial overlap index: scientific reports” [10], with a total of 656 citations, while the article titled “Development and implementation of intraoperative magnetic resonance imaging and its neurosurgical applications”, authored by Black et al., was the most locally cited article related to IOMRI [11] (Appendix A). This article has garnered 49 local and 577 global citations, with 8.49% of all its citations accrued locally (Appendix A). It is of note that the review article titled “Interventional MR imaging: concepts, systems, and applications in neuroradiology” by Lewin JS [12] has the highest local citation percentage of 15.69%.

## 4. Discussion

Bibliometric analyses can help provide insight into the status of research within a particular field, identifying strengths of research and areas of lacunae. Moreover, articles that can assist researchers, trainees, and clinicians can be highlighted [13,14]. To the best of the authors’ knowledge, this is the first instance where bibliometric analyses have been used to identify the most significant studies, individuals, institutions, and research disciplines, with respect to the use of IOMRI. By identifying the ‘Top 100′ articles in this field (Table 5), readers are directed to the most relevant articles.

These articles were published in 138 different journals, the majority being journals of neurosurgery, (neuro)oncology, and radiology. This is explained by the uniqueness of this field, which involves the collaboration of these two branches (Figure 6A,B). This is further exemplified in the top two most cited articles. The most cited article (globally) by Zou et al. [10] elaborates on the science behind IOMRI and deals with the validation of image segmentation, based on a spatial overlap index. This index, the Dice similarity coefficient (DSC), was validated on two sets of patients, the first group being 10 consecutive patients who underwent prostatic brachytherapy and were imaged with a 1.5 T preoperative MRI and a 0.5 Tesla MRI, and a second group of patients were 9 patients with 3 types of brain tumours. DSC values were computed, and logit-transformed values were compared in the mean with the analysis of variance (ANOVA). The article concludes that the DSC value is a useful, yet simple, tool that, when used for the analyses of spatial overlap, can be used to ensure reproducibility and accuracy in image segmentation. The validity of this approach paved the way for further studies, ensuring the fidelity of MRI image analysis, which has led to an average of 38.59 citations per year since 2004.

Another article with a large number of citations per year was authored by Senft et al. This study constituted a randomised controlled trial, assessing the extent of resection in glioma surgery using IOMRI [15]. The authors reported that a higher number of patients that underwent IOMRI guided surgery had complete resection of the tumour (23 of 24 patients/96%) than patients in the control group, where conventional surgery was performed (17 of 25/68% with a *p* value of 0.023). Postoperative rates of new neurological deficits did not significantly or statistically differ between the two groups (3 patients (13%) in the IOMRI group and 2 (8%) patients in the control group). This article conclusively establishes the efficacy and superiority of IOMRI, explaining the number of citations. On a deeper analysis, this article emerged as having the highest number of citations per year (40.30), highlighting that, when a simple metric such as ‘total citations’ alone are utilised, high impact articles may be looked over. It also indicates that randomised controlled trials continue to have the highest impact among all study types. Additionally, the article by Black et al. [11], which describes the first IOMRI to be utilised using a 0.5 T intraoperative system, has the highest local citations and the second highest global citation rate. This is due to the pioneering nature of the article and its longevity.

An interesting aspect of this bibliometric analysis is that IOMRI is a recent and, therefore, nascent advancement in neurosurgery. Therefore, although Percival Bailey and Harvey Cushing introduced the term Glioma in 1926 [16], the articles on IOMRI primarily focussed on improved glioma resection, using IOMRI dates from 1996 to 2016. This makes the analysis unique, as most other areas in neurosurgery are far older and IOMRI is a new, growing field, and this analysis enlightens areas of relevant research [17]. Fields such as cardiology and thrombolysis, which have made rapid strides in recent years, have shown similar results [18].

These articles were predominantly published in journals dealing with three specialities: namely, Neurosciences, Radiology, and Oncology, with 95 (of 100) articles being in these three domains. Indicating the degree of collaboration required in the development, utilisation, and outcomes in IOMRI. Data garnered and analysed, in the course of preparing this list of top cited articles, also showed preponderance of articles and consistent source growth in newer journals such as World Neurosurgery (est. 2010), indicating continued research and a general increase in the focus on IOMRI. In time, it is feasible that these articles may garner additional citations, adding to their growing relevance.

Maximum number of articles originated from the United States, followed by Germany (Figure 7A,B). This differs from other bibliometric analyses, where the United States has significantly larger impact indices than other countries. This is probably owing to the fact that the US spends nearly three times the amount of government funds as countries in Europe for research [19]. The probable reason for Germany’s impact in the field is the near simultaneous establishment of a 0.2 T MRI with a patient transportation system by the Erlangen and Heidelberg groups [20]. Prior to the establishment of IOMRI, their group demonstrated the benefits and emphasised the role of early post-operative imaging [21], which then translated to a series encompassing 47 patients with gliomas, wherein they documented additional resection of overlooked or microsurgically non identifiable tumour (Figure 8) remnants using IOMRI [22]. Owing to the degree of technological research and funding involved in the research and utilisation of IOMRI, the contribution of low-income countries has been minimal.

The dominance of Germany is further exemplified when considering institutional affiliations (of the considered articles), where the Friedrich-Alexander-Universität (FAU) Erlangen-Nurnberg, founded in 1743, is the largest contributor for these high impact articles. In addition to the Cluster of Excellence, ‘Engineering of Advanced Materials’ (EAM), and the ‘Graduate School of Advanced Optical Technologies’ (SAOT), which was founded as part of the Excellence Initiative, FAU currently has more than 40 co-ordinated programmes funded by the German Research Foundation [23]. This is closely followed by the University of Minnesota and Harvard University, which are, again, among the top 50 NIH funded institutions [24]. Furthermore, NIH funding for brain tumours, and by extension, IOMRI, has increased over the years, which may be explained by the higher incidence of brain tumours among developed nations [25,26]. This may also be due to the fact that other diseases, which are more prevalent, such as diabetes mellitus and hypertension have well established and validated guidelines and, therefore, require lesser allocation of funds towards research (Figure 9).

It has been nearly three decades since the first IOMRI was established, and its defining role and utility for intraoperative imaging, in glioma surgery and beyond, is widely accepted and established. The permeant advancements of MR technologies, for the detection of brain function, tumour borders, and brain metabolism, ensure that this neurosurgical tool will remain superior to other technologies (such as optical imaging, ultrasound, etc.) for safe and radical tumour removal.

Bibliometric analysis is a purely quantitative tool which measures the impact as perceived by the scientific community and not a representation of the ‘real world’ impact of research. Therefore, while these articles may have a high citation rate, their role in changing patient care may be entirely different. In addition, as only a single database was used for analysis, bias may be encountered. Although certain articles exclude self-citations, they were not ruled out in this study, as the authors believe that only continued work in a particular field leads to self-citations and that this is a marker of a focussed researcher rather than one who continually changes areas of research [8].

### Current Limitations and Future Areas of Interest

This article highlights avenues of research in IOMRI that may be more relevant than others. Ergonomic and economically viable methods of IOMRI will increase its availability and thereby, its use in Asian regions. The initial cost in procuring and subsequent economic factors in utilising an IOMRI have erected barriers in its widespread adoption, while ergonomic factors, the learning curve associated with the introduction and familiarisation of the operative team, have led to compliance issues among surgeons with access to IOMRI facilities. Furthermore, in an era of ever advancing technologies, failure to address ergonomic issues will lead to an ever shrinking, potentially cluttered operating room where surgical teams are forced to accommodate shrinking workspaces. Addressing these limitations will dramatically increase the number of patients treated using IOMRI and constitute a higher relevance. Additionally, increased funding of research, or for patient care via insurance companies, will increase the availability of these advanced technologies, benefitting the population overall. It is interesting to note that posterior fossa lesions, spinal tumours, and paediatric populations did not feature in these articles, indicating that these may be areas that warrant additional and future research.

One glaring limitation of bibliometric analysis is that the number of citations for each paper will increase over the years [4]. Thus, the number of citations of the articles published recently may be small despite having a greater impact. The “Average years from publication” in Table 1 also suggests this point. This means that the analysis of the top 100 most cited papers may exclude the latest-trend papers unintentionally and selectively. However, this only reiterates the need of periodic analysis to identify the trends.

## 5. Conclusions

This bibliometric analysis shows that most research on IOMRI originated from developed countries such as the USA and Germany. It provides insights into the evolution, utilisation, and reference points for further studies on IOMRI, highlighting ‘blind spots’ in research, such as the exclusion of paediatric populations, spinal tumours, and posterior fossa pathologies. The need for additional funding, especially in the developing world, and better ergonomics has been established. Randomised control trials and focused research on a singular topic have been shown to increase impact. We hope this analysis will lead to more areas being explored in this nascent, yet powerful, technique.

## Figures and Tables

**Figure 1 life-12-00175-f001:**
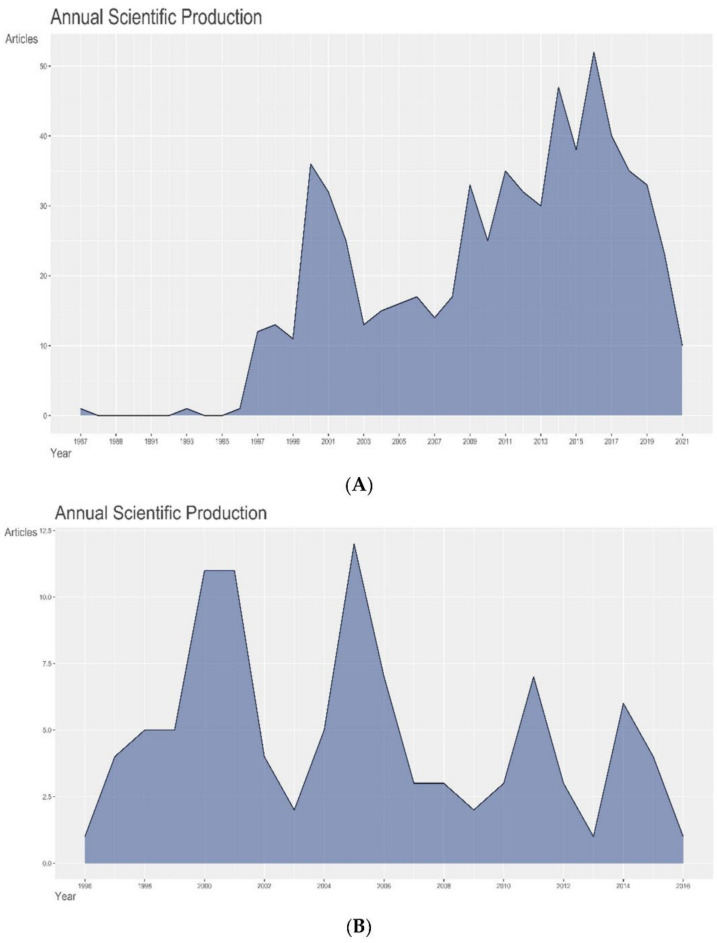
Graph demonstrating (**A**) annual production of ALL articles on intraoperative MRI, (**B**) annual production of the 100 most cited articles on intraoperative MRI, and (**C**) average article citations per year of the 100 most cited articles on intraoperative MRI.

**Figure 2 life-12-00175-f002:**
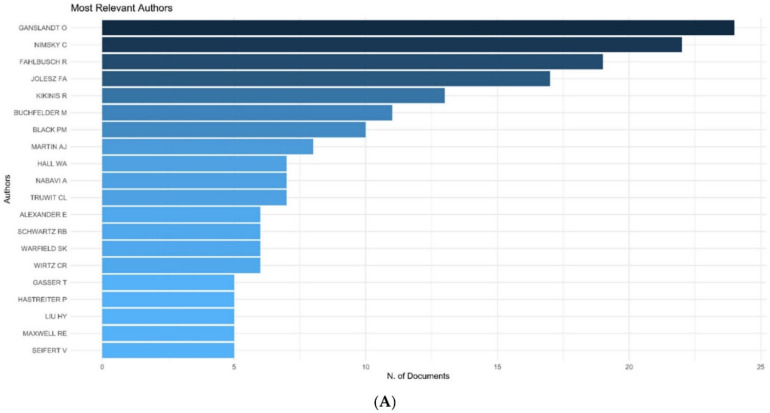
Graph demonstrating (**A**) authors with maximum number of publications in the top 100 cited articles, (**B**) author impact in terms of h-index among all the articles published on IOMRI, and (**C**) production rate of the authors per year of the 100 most cited articles on intraoperative MRI.

**Figure 3 life-12-00175-f003:**
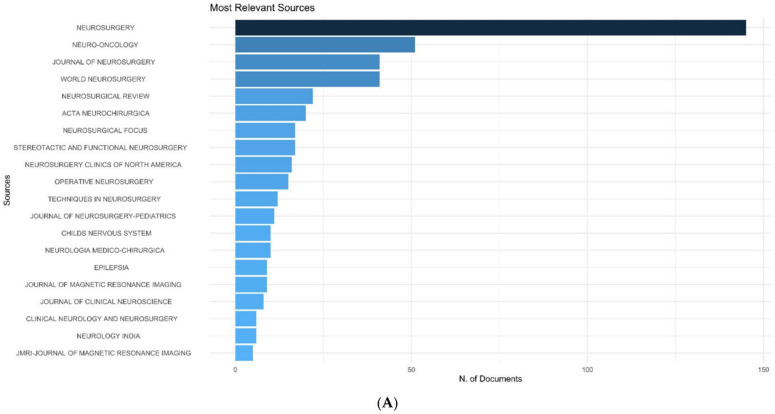
Graph demonstrating (**A**) most relevant source journals, which published the articles on intraoperative MRI, (**B**) Bradford’s law related to journal sources of the 100 most cited articles on intraoperative MRI.

**Figure 4 life-12-00175-f004:**
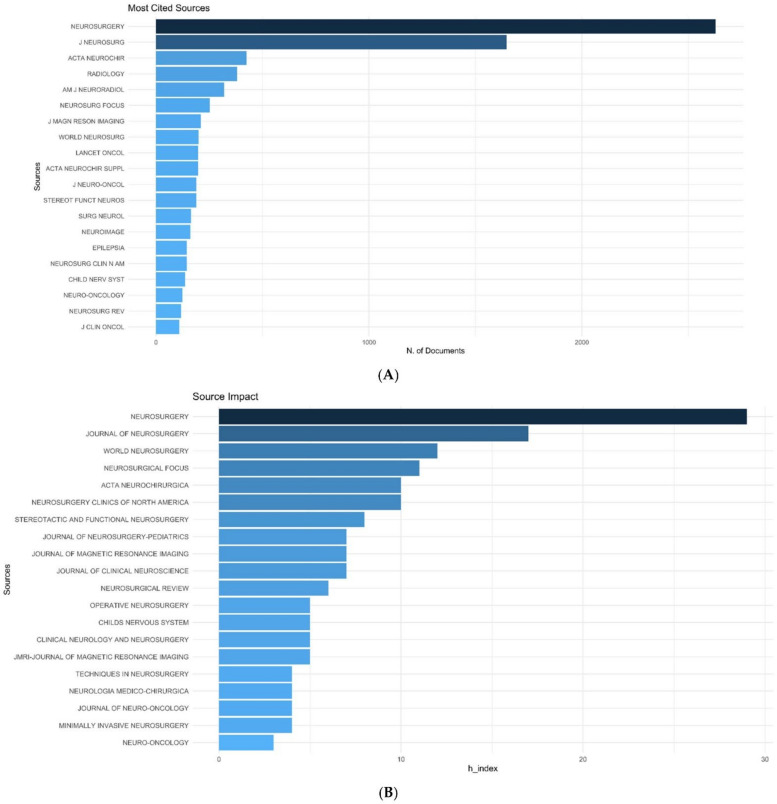
Graph demonstrating (**A**) the most cited source journals which published the articles on intraoperative MRI, (**B**) source impact, in terms of h-index, among all the articles published on IOMRI, and (**C**) source dynamics, showing the cumulative number of publications in the different journals over time.

**Figure 5 life-12-00175-f005:**
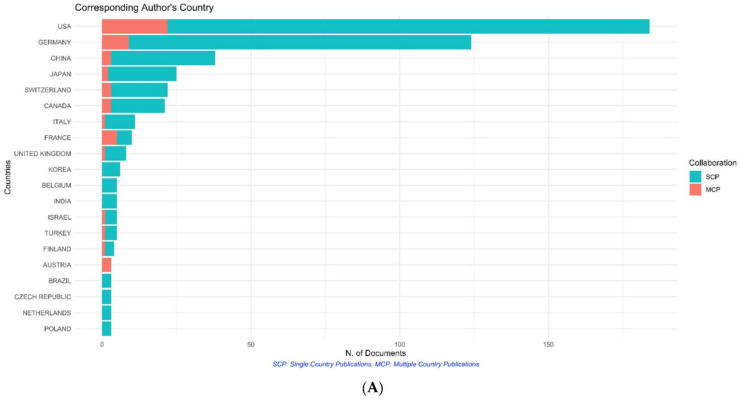
Graph demonstrating the (**A**) origin country of the corresponding authors of all the articles published on IOMRI, (**B**) origin country of the corresponding authors of the top 100 cited articles published on IOMRI, (**C**) most cited countries, and (**D**) most relevant affiliations of all the articles on intraoperative MRI.

**Figure 6 life-12-00175-f006:**
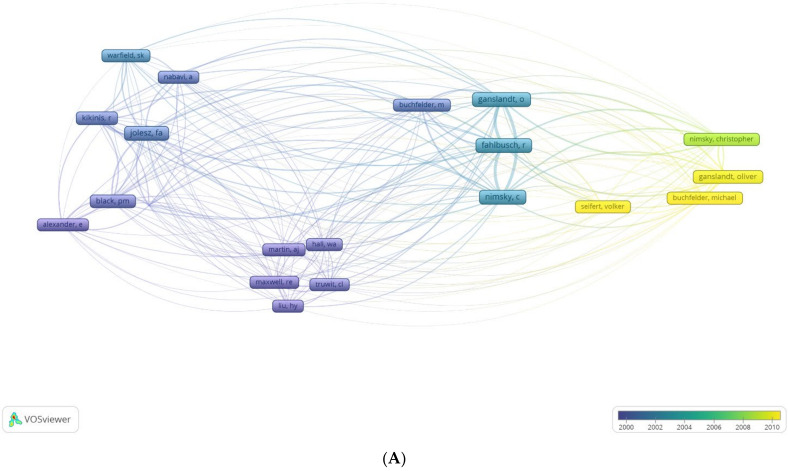
A VOS (visualising of similarities) graph of bibliometric analysis, of (**A**) most cited authors and (**B**) relative influence of these authors, of the 100 most cited articles on intraoperative MRI. The density graph represents the same, with the authors with more impact highlighted.

**Figure 7 life-12-00175-f007:**
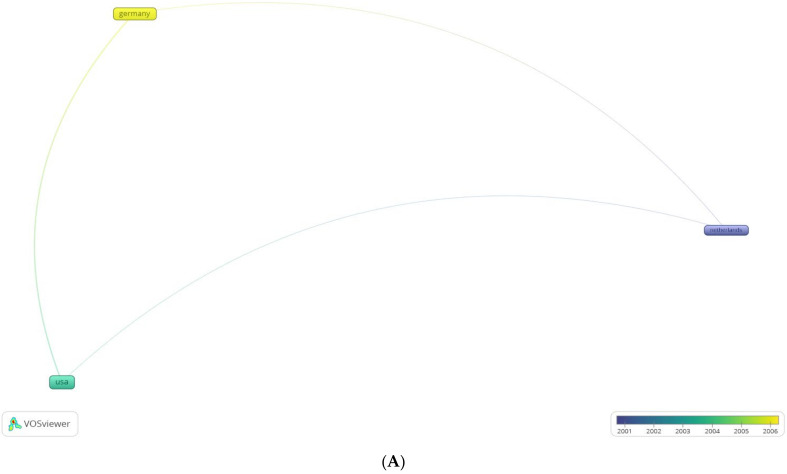
A VOS (visualising of similarities) graph of bibliometric analysis of (**A**) most cited nations with originating research, with overlay graph of co-citation coupling of documents, and (**B**) relative influence of the most cited institutions of the 100 most cited articles on intraoperative MRI.

**Figure 8 life-12-00175-f008:**
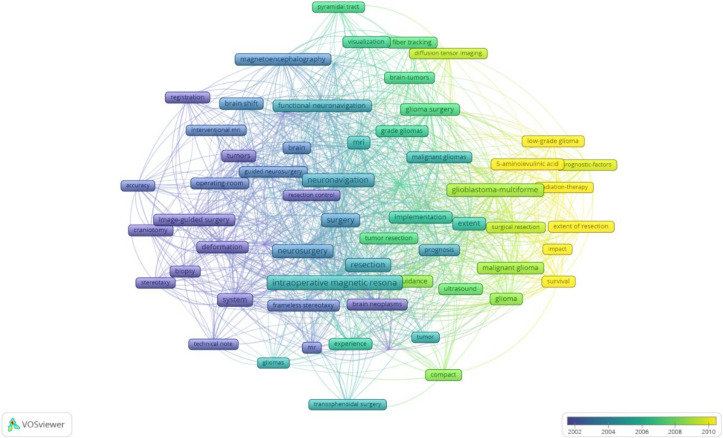
A VOS (visualising of similarities) graph of bibliometric analysis of the MRI and the overlay graph of co-occurrence of keywords in the 100 most cited articles on intraoperative MRI.

**Figure 9 life-12-00175-f009:**
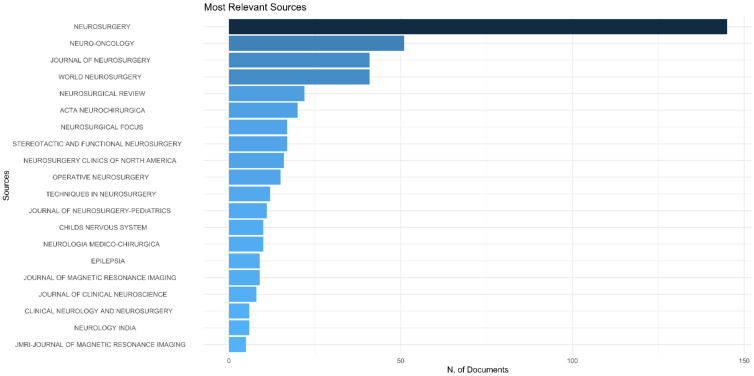
Figure showing the most frequently used keywords.

**Table 1 life-12-00175-t001:** Analysis of the major findings from the bibliometric analysis of the 100 most cited papers with intraoperative MRI.

Description	All Articles	Top 100 Cited Articles
Timespan	1996:2020	1996:2020
Sources (Journals, Books, etc.)	138	42
Documents	663	100
Average years from publication	10.5	16
Average citations per documents	17.52	113.5
Average citations per year per doc	1.451	7.444
References	6438	2089
**DOCUMENT TYPES**		
Article	367	76
Article; proceedings paper	17	6
Reprint	1	1
Review	31	14
Others (Editorials/Book Chapters/Meeting abstract etc.)	247	--
**DOCUMENT CONTENTS**		
Keywords Plus (ID)	848	319
Author’s Keywords (DE)	755	203
**AUTHORS**		
Authors	1984	350
Author Appearances	3740	613
Authors of single-authored documents	51	3
Authors of multi-authored documents	1933	347
**AUTHORS COLLABORATION**		
Single-authored documents	76	4
Documents per Author	0.334	0.286
Authors per Document	2.99	3.5
Co-Authors per Documents	5.64	6.13

**Table 2 life-12-00175-t002:** A review of the bibliometrics for the most influential authors in intraoperative MRI related research.

Author	H Index	G Index	M Index	Total Citations	Number of Articles	Year
NIMSKY C	23	47	0.958	2218	51	1998
FAHLBUSCH R	22	45	0.88	2026	46	1997
BUCHFELDER M	18	31		1385	31	1998
GANSLANDT O	22	29	0.917	2175	29	1998
BLACK PM	13	27	0.5	1896	27	1996
SCHULDER M	7	13		172	25	1997
WIRTZ CR	14	25	0.56	921	25	1997
JOLESZ FA	14	23	0.538	1951	23	1996
SUTHERLAND GR	9	16	0.391	271	22	1999
COBURGER J	11	19	1.375	368	19	2014
SEIFERT V	10	18	0.417	852	18	1998
CHEN XL	8	14	0.727	197	17	2011
TRONNIER VM	6	17	0.24	527	17	1997
CHICOINE MR	6	11	0.462	128	16	2009
SENFT C	7	16	0.5	747	16	2008
XU BN	8	14	0.727	211	16	2011
GASSER T	8	13	0.471	790	13	2005
NABAVI A	8	13	0.348	529	13	1999
EVANS J	2	6	0.154	43	12	2009
KONIG R	8	11	0.8	312	11	2012
PRABHU SS	6	11	0.462	303	11	2009
ROESSLER K	5	10		111	11	2009
SAMII A	6	10	0.545	109	11	2011
SOMMER B	5	10	0.5	115	11	2012
BERNAYS RL	7	10	0.28	235	10	1997
DACEY RG	4	10	0.308	112	10	2009
RAO G	7	10	0.5	302	10	2008
YU XG	4	9	0.364	91	10	2011
HLAVAC M	6	9	0.6	171	9	2012
LEUTHARDT EC	2	6	0.154	38	9	2009
REGLI L	4	5		37	9	2016
SCHWARTZ RB	6	9	0.24	1364	9	1997
WEINBERG JS	6	9	0.333	279	9	2004
ALEXANDER E	6	8	0.231	1087	8	1996
BARNETT GH	5	8	0.25	183	8	2002
BOZINOV O	3	5		33	8	2016
FRANZ K	5	8	0.357	650	8	2008
GIORDANO M	5	8	0.455	88	8	2011
HALL WA	2	6	0.091	43	8	2000
HAMER HM	4	8	0.4	104	8	2012
JENSEN RL	1	2	0.25	6	8	2018
KIKINIS R	6	8	0.231	1265	8	1996
MURAGAKI Y	4	8	0.235	82	8	2005
REYNS N	3	4		21	8	2017
RICH KM	3	8	0.231	75	8	2009
RODER C	5	8	0.556	128	8	2013
SAWAYA R	5	8	0.278	225	8	2004
SMYTH MD	2	6	0.154	43	8	2009
SUN GC	6	8	0.545	126	8	2011
TATAGIBA M	4	8	0.4	112	8	2012

**Table 3 life-12-00175-t003:** The most influential journals with intraoperative MRI related research.

Sources	Articles
NEUROSURGERY	145
NEURO-ONCOLOGY	51
JOURNAL OF NEUROSURGERY	41
WORLD NEUROSURGERY	41
NEUROSURGICAL REVIEW	22
ACTA NEUROCHIRURGICA	20
NEUROSURGICAL FOCUS	17
STEREOTACTIC AND FUNCTIONAL NEUROSURGERY	17
NEUROSURGERY CLINICS OF NORTH AMERICA	16
OPERATIVE NEUROSURGERY	15
TECHNIQUES IN NEUROSURGERY	12
JOURNAL OF NEUROSURGERY-PEDIATRICS	11
CHILDS NERVOUS SYSTEM	10
NEUROLOGIA MEDICO-CHIRURGICA	10
EPILEPSIA	9
JOURNAL OF MAGNETIC RESONANCE IMAGING	9
JOURNAL OF CLINICAL NEUROSCIENCE	8
CLINICAL NEUROLOGY AND NEUROSURGERY	6
NEUROLOGY INDIA	6
JMRI-JOURNAL OF MAGNETIC RESONANCE IMAGING	5
JOURNAL OF NEURO-ONCOLOGY	5
MINIMALLY INVASIVE NEUROSURGERY	5
NEUROCHIRURGIE	5
ANESTHESIA AND ANALGESIA	4
BRITISH JOURNAL OF NEUROSURGERY	4
JOURNAL OF NEUROLOGICAL SURGERY PART A-CENTRAL EUROPEAN NEUROSURGERY	4
JOURNAL OF NEUROSURGICAL ANESTHESIOLOGY	4
MOVEMENT DISORDERS	4
PEDIATRIC NEUROSURGERY	4
RADIOLOGY	4
CELL TRANSPLANTATION	3
CURRENT OPINION IN ANESTHESIOLOGY	3
EUROPEAN RADIOLOGY	3
LANCET ONCOLOGY	3
NEUROIMAGE	3
NEUROLOGICAL RESEARCH	3
NEUROLOGICAL SURGERY	3
NEUROLOGY	3
NEURORADIOLOGY	3
PITUITARY	3
SURGICAL NEUROLOGY	3
TURKISH NEUROSURGERY	3
ACADEMIC RADIOLOGY	2
AMERICAN JOURNAL OF ROENTGENOLOGY	2
ANESTHESIOLOGY	2
AORN JOURNAL	2
CHINESE MEDICAL JOURNAL	2
EXPERIMENTAL AND CLINICAL ENDOCRINOLOGY & DIABETES	2
FRONTIERS IN ONCOLOGY	2
INTERNATIONAL JOURNAL OF COMPUTER ASSISTED RADIOLOGY AND SURGERY	2
MINIMALLY INVASIVE THERAPY & ALLIED TECHNOLOGIES	2
PEDIATRIC BLOOD & CANCER	2
PEDIATRIC RADIOLOGY	2
PHOTODIAGNOSIS AND PHOTODYNAMIC THERAPY	2
RADIOLOGE	2
ROFO-FORTSCHRITTE AUF DEM GEBIET DER RONTGENSTRAHLEN UND DER BILDGEBENDEN VERFAHREN	2
SCIENTIFIC REPORTS	2
SPINE	2
SWISS MEDICAL WEEKLY	2
ACTA RADIOLOGICA	1
ADVANCES IN CLINICAL AND EXPERIMENTAL MEDICINE	1
AMERICAN JOURNAL OF NEURORADIOLOGY	1
AMERICAN JOURNAL OF RHINOLOGY	1
ANAESTHESIA AND INTENSIVE CARE	1
ANATOMICAL SCIENCES EDUCATION	1
ANNALS OF NEUROLOGY	1
ANNALS OF ONCOLOGY	1
ARQUIVOS DE NEURO-PSIQUIATRIA	1
ASIAN JOURNAL OF SURGERY	1
BOSNIAN JOURNAL OF BASIC MEDICAL SCIENCES	1
BREAST CANCER RESEARCH AND TREATMENT	1
BRITISH JOURNAL OF ANAESTHESIA	1
CANADIAN JOURNAL OF ANAESTHESIA-JOURNAL CANADIEN D ANESTHESIE	1
CANADIAN MEDICAL ASSOCIATION JOURNAL	1
CANCER	1
CANCER MANAGEMENT AND RESEARCH	1
CANCERS	1
CESKA A SLOVENSKA NEUROLOGIE A NEUROCHIRURGIE	1
CLINICAL NEURORADIOLOGY	1
CLINICAL RADIOLOGY	1
CRITICAL REVIEWS IN NEUROSURGERY	1
CURRENT MEDICAL IMAGING	1
CURRENT MEDICAL RESEARCH AND OPINION	1
CURRENT OPINION IN NEUROLOGY	1
EJC SUPPLEMENTS	1
ELECTROENCEPHALOGRAPHY AND CLINICAL NEUROPHYSIOLOGY	1
ENDOCRINE	1
ENDOCRINE CONNECTIONS	1
EUROPEAN JOURNAL OF CANCER	1
EUROPEAN JOURNAL OF ENDOCRINOLOGY	1
EUROPEAN JOURNAL OF NEUROLOGY	1
EUROPEAN JOURNAL OF RADIOLOGY	1
EXPERT REVIEW OF ANTICANCER THERAPY	1
EXPERT REVIEW OF NEUROTHERAPEUTICS	1
EXPLORE-THE JOURNAL OF SCIENCE AND HEALING	1
HNO	1
IEEE ROBOTICS AND AUTOMATION LETTERS	1
IEEE TRANSACTIONS ON MEDICAL IMAGING	1
IMAGING IN ENDOCRINE DISORDERS	1
INTERNATIONAL JOURNAL OF CLINICAL AND EXPERIMENTAL MEDICINE	1
INTERNATIONAL JOURNAL OF RADIATION ONCOLOGY BIOLOGY PHYSICS	1
JOURNAL OF CEREBRAL BLOOD FLOW AND METABOLISM	1
JOURNAL OF CLINICAL ANESTHESIA	1
JOURNAL OF CLINICAL MONITORING AND COMPUTING	1
JOURNAL OF COMPUTER ASSISTED TOMOGRAPHY	1
JOURNAL OF NEUROLOGICAL SCIENCES-TURKISH	1
JOURNAL OF NEUROLOGICAL SURGERY PART B-SKULL BASE	1
JOURNAL OF NEUROLOGY	1
JOURNAL OF NEURORADIOLOGY	1
JOURNAL OF NEUROSURGERY-SPINE	1
JOURNAL OF NEUROSURGICAL SCIENCES	1
JOURNAL OF SURGICAL ONCOLOGY	1
JOURNAL OF THE NEUROLOGICAL SCIENCES	1
JOURNAL OF UROLOGY	1
KNEE SURGERY SPORTS TRAUMATOLOGY ARTHROSCOPY	1
LARYNGOSCOPE	1
MAGNETIC RESONANCE IMAGING	1
MAGNETIC RESONANCE IN MEDICINE	1
MAYO CLINIC PROCEEDINGS	1
MEDICAL IMAGE COMPUTING AND COMPUTER-ASSISTED INTERVENTION MICCAI’99 PROCEEDINGS	1
MOLECULAR THERAPY	1
NAGOYA JOURNAL OF MEDICAL SCIENCE	1
NEUROCIRUGIA	1
NEUROIMAGE-CLINICAL	1
NEUROIMAGING CLINICS OF NORTH AMERICA	1
NEUROLOGIA I NEUROCHIRURGIA POLSKA	1
NEUROSURGERY QUARTERLY	1
ONCOLOGY LETTERS	1
PEDIATRIC ANESTHESIA	1
PLOS ONE	1
SEMINARS IN INTERVENTIONAL RADIOLOGY	1
SKULL BASE-AN INTERDISCIPLINARY APPROACH	1
SMALL	1
STROKE	1
SURGICAL LAPAROSCOPY ENDOSCOPY & PERCUTANEOUS TECHNIQUES	1
TECHNOLOGY IN CANCER RESEARCH & TREATMENT	1
UNFALLCHIRURG	1
WORLD JOURNAL OF SURGICAL ONCOLOGY	1

**Table 4 life-12-00175-t004:** Bibliometrics of the top impactful journals with Intraoperative MRI related research.

Source	H Index	G Index	M Index	Total Citations	Total Number of Articles	First Article
NEUROSURGERY	29	64	0.82857143	4189	145	1987
JOURNAL OF NEUROSURGERY	17	28	0.80952381	833	41	2001
WORLD NEUROSURGERY	12	18	1	377	41	2010
NEUROSURGICAL FOCUS	11	17	0.91666667	314	17	2010
ACTA NEUROCHIRURGICA	10	16	0.5	277	20	2002
NEUROSURGERY CLINICS OF NORTH AMERICA	10	16	0.38461539	366	16	1996
STEREOTACTIC AND FUNCTIONAL NEUROSURGERY	8	15	0.32	227	17	1997
JOURNAL OF NEUROSURGERY-PEDIATRICS	7	11	0.53846154	150	11	2009
JOURNAL OF MAGNETIC RESONANCE IMAGING	7	9	0.31818182	208	9	2000
JOURNAL OF CLINICAL NEUROSCIENCE	7	8	0.33333333	168	8	2001
NEUROSURGICAL REVIEW	6	12		145	22	1998
OPERATIVE NEUROSURGERY	5	6	0.625	42	15	2014
CHILDS NERVOUS SYSTEM	5	10	0.22727273	116	10	2000
CLINICAL NEUROLOGY AND NEUROSURGERY	5	6	0.41666667	128	6	2010
JMRI-JOURNAL OF MAGNETIC RESONANCE IMAGING	5	5	0.20833333	183	5	1998
TECHNIQUES IN NEUROSURGERY	4	5	0.18181818	34	12	2000
NEUROLOGIA MEDICO-CHIRURGICA	4	8	0.22222222	65	10	2004
JOURNAL OF NEURO-ONCOLOGY	4	5	0.22222222	109	5	2004
MINIMALLY INVASIVE NEUROSURGERY	4	5	0.21052632	134	5	2003
NEURO-ONCOLOGY	3	15	0.16666667	239	51	2004
NEUROCHIRURGIE	3	5	0.21428571	36	5	2008
BRITISH JOURNAL OF NEUROSURGERY	3	4		26	4	2015
JOURNAL OF NEUROSURGICAL ANESTHESIOLOGY	3	4	0.15789474	61	4	2003
PEDIATRIC NEUROSURGERY	3	4	0.14285714	63	4	2001
CURRENT OPINION IN ANESTHESIOLOGY	3	3	0.27272727	28	3	2011
LANCET ONCOLOGY	3	3	0.27272727	670	3	2011
NEUROIMAGE	3	3	0.17647059	114	3	2005
EPILEPSIA	2	8	0.08695652	79	9	1999
NEUROLOGY INDIA	2	4	0.10526316	19	6	2003
ANESTHESIA AND ANALGESIA	2	3	0.11111111	11	4	2004
JOURNAL OF NEUROLOGICAL SURGERY PART A-CENTRAL EUROPEAN NEUROSURGERY	2	3	0.22222222	13	4	2013
EUROPEAN RADIOLOGY	2	3	0.1	95	3	2002
NEURORADIOLOGY	2	3		124	3	2004
PITUITARY	2	3		43	3	2015
TURKISH NEUROSURGERY	2	3	0.2	13	3	2012
ACADEMIC RADIOLOGY	2	2	0.11764706	79	2	2005
MINIMALLY INVASIVE THERAPY & ALLIED TECHNOLOGIES	2	2	0.09090909	30	2	2000
PEDIATRIC RADIOLOGY	2	2	0.2	29	2	2012
PHOTODIAGNOSIS AND PHOTODYNAMIC THERAPY	2	2	0.22222222	80	2	2013
RADIOLOGE	2	2	0.08333333	49	2	1998
RADIOLOGY	1	2	0.04166667	4	4	1998
CELL TRANSPLANTATION	1	2	0.2	5	3	2017
NEUROLOGICAL RESEARCH	1	3		145	3	2006
NEUROLOGICAL SURGERY	1	2	0.04347826	4	3	1999
SURGICAL NEUROLOGY	1	3	0.04761905	9	3	2001
AMERICAN JOURNAL OF ROENTGENOLOGY	1	2	0.06666667	19	2	2007
ANESTHESIOLOGY	1	1	0.03448276	1	2	1993
AORN JOURNAL	1	2	0.11111111	9	2	2013
CHINESE MEDICAL JOURNAL	1	2	0.1	8	2	2012
FRONTIERS IN ONCOLOGY	1	2	0.25	5	2	2018
INTERNATIONAL JOURNAL OF COMPUTER ASSISTED RADIOLOGY AND SURGERY	1	1	0.0625	1	2	2006
ROFO-FORTSCHRITTE AUF DEM GEBIET DER RONTGENSTRAHLEN UND DER BILDGEBENDEN VERFAHREN	1	1	0.1	2	2	2012
SCIENTIFIC REPORTS	1	2	0.14285714	10	2	2015
SPINE	1	2	0.04761905	37	2	2001
SWISS MEDICAL WEEKLY	1	2	0.1	13	2	2012
ACTA RADIOLOGICA	1	1	0.06666667	5	1	2007
ADVANCES IN CLINICAL AND EXPERIMENTAL MEDICINE	1	1	0.1	11	1	2012
AMERICAN JOURNAL OF NEURORADIOLOGY	1	1	0.125	35	1	2014
AMERICAN JOURNAL OF RHINOLOGY	1	1	0.0625	32	1	2006
ANAESTHESIA AND INTENSIVE CARE	1	1	0.11111111	1	1	2013
ANATOMICAL SCIENCES EDUCATION	1	1	0.11111111	3	1	2013
ANNALS OF ONCOLOGY	1	1	0.33333333	2	1	2019
ARQUIVOS DE NEURO-PSIQUIATRIA	1	1	0.09090909	4	1	2011
ASIAN JOURNAL OF SURGERY	1	1	0.14285714	7	1	2015
BOSNIAN JOURNAL OF BASIC MEDICAL SCIENCES	1	1	0.33333333	1	1	2019
BRITISH JOURNAL OF ANAESTHESIA	1	1	0.1	1	1	2012
CANADIAN JOURNAL OF ANAESTHESIA-JOURNAL CANADIEN D ANESTHESIE	1	1	0.05	15	1	2002
CANADIAN MEDICAL ASSOCIATION JOURNAL	1	1	0.04347826	17	1	1999
CANCER	1	1	0.05882353	319	1	2005
CESKA A SLOVENSKA NEUROLOGIE A NEUROCHIRURGIE	1	1	0.07692308	3	1	2009
CLINICAL NEURORADIOLOGY	1	1	0.25	1	1	2018
CLINICAL RADIOLOGY	1	1	0.16666667	5	1	2016
CRITICAL REVIEWS IN NEUROSURGERY	1	1	0.04347826	1	1	1999
CURRENT MEDICAL RESEARCH AND OPINION	1	1	0.2	20	1	2017
CURRENT OPINION IN NEUROLOGY	1	1	0.05555556	25	1	2004
ELECTROENCEPHALOGRAPHY AND CLINICAL NEUROPHYSIOLOGY	1	1	0.04166667	2	1	1998
ENDOCRINE	1	1	0.1	36	1	2012
ENDOCRINE CONNECTIONS	1	1	0.25	3	1	2018
EUROPEAN JOURNAL OF ENDOCRINOLOGY	1	1	0.05882353	77	1	2005
EUROPEAN JOURNAL OF RADIOLOGY	1	1	0.05882353	40	1	2005
EXPERT REVIEW OF ANTICANCER THERAPY	1	1	0.07692308	13	1	2009
EXPERT REVIEW OF NEUROTHERAPEUTICS	1	1	0.08333333	12	1	2010
HNO	1	1	0.2	5	1	2017
IEEE ROBOTICS AND AUTOMATION LETTERS	1	1	0.25	12	1	2018
IEEE TRANSACTIONS ON MEDICAL IMAGING	1	1	0.05882353	162	1	2005
IMAGING IN ENDOCRINE DISORDERS	1	1	0.16666667	8	1	2016
INTERNATIONAL JOURNAL OF CLINICAL AND EXPERIMENTAL MEDICINE	1	1	0.14285714	5	1	2015
JOURNAL OF CLINICAL ANESTHESIA	1	1	0.07692308	17	1	2009
JOURNAL OF CLINICAL MONITORING AND COMPUTING	1	1	0.1	1	1	2012
JOURNAL OF COMPUTER ASSISTED TOMOGRAPHY	1	1	0.04	64	1	1997
JOURNAL OF NEUROLOGY	1	1	0.16666667	26	1	2016
JOURNAL OF NEURORADIOLOGY	1	1	1	1	1	2021
JOURNAL OF NEUROSURGERY-SPINE	1	1	0.07142857	14	1	2008
JOURNAL OF NEUROSURGICAL SCIENCES	1	1	0.2	17	1	2017
JOURNAL OF SURGICAL ONCOLOGY	1	1	0.08333333	19	1	2010
JOURNAL OF UROLOGY	1	1	0.2	1	1	2017
KNEE SURGERY SPORTS TRAUMATOLOGY ARTHROSCOPY	1	1	0.2	17	1	2017
LARYNGOSCOPE	1	1	0.04761905	24	1	2001
MAGNETIC RESONANCE IMAGING	1	1	0.11111111	7	1	2013
MAGNETIC RESONANCE IN MEDICINE	1	1	0.2	3	1	2017
MAYO CLINIC PROCEEDINGS	1	1	0.14285714	38	1	2015
MEDICAL IMAGE COMPUTING AND COMPUTER-ASSISTED INTERVENTION, MICCAI’99, PROCEEDINGS	1	1	0.04347826	11	1	1999
NAGOYA JOURNAL OF MEDICAL SCIENCE	1	1	0.07692308	4	1	2009
NEUROCIRUGIA	1	1	0.11111111	5	1	2013
NEUROIMAGE-CLINICAL	1	1	0.11111111	15	1	2013
NEUROIMAGING CLINICS OF NORTH AMERICA	1	1	0.05	22	1	2002
NEUROLOGIA I NEUROCHIRURGIA POLSKA	1	1	0.09090909	11	1	2011
NEUROSURGERY QUARTERLY	1	1	0.05263158	5	1	2003
ONCOLOGY LETTERS	1	1	0.14285714	1	1	2015
PEDIATRIC ANESTHESIA	1	1	0.09090909	4	1	2011
PLOS ONE	1	1	0.14285714	54	1	2015
SKULL BASE-AN INTERDISCIPLINARY APPROACH	1	1	0.09090909	23	1	2011
SMALL	1	1	0.14285714	67	1	2015
TECHNOLOGY IN CANCER RESEARCH & TREATMENT	1	1	0.07692308	14	1	2009
UNFALLCHIRURG	1	1	0.1	6	1	2012
WORLD JOURNAL OF SURGICAL ONCOLOGY	1	1	0.14285714	4	1	2015
MOVEMENT DISORDERS	0	0	0	0	4	2013
NEUROLOGY	0	0	0	0	3	2016
EXPERIMENTAL AND CLINICAL ENDOCRINOLOGY & DIABETES	0	0	0	0	2	2007
PEDIATRIC BLOOD & CANCER	0	0	0	0	2	2014
ANNALS OF NEUROLOGY	0	0	0	0	1	2019
BREAST CANCER RESEARCH AND TREATMENT	0	0	0	0	1	2002
CANCER MANAGEMENT AND RESEARCH	0	0	0	0	1	2018
CANCERS	0	0	0	0	1	2020
CURRENT MEDICAL IMAGING	0	0	0	0	1	2010
EJC SUPPLEMENTS	0	0	0	0	1	2009
EUROPEAN JOURNAL OF CANCER	0	0	0	0	1	2013
EUROPEAN JOURNAL OF NEUROLOGY	0	0	0	0	1	2008
EXPLORE-THE JOURNAL OF SCIENCE AND HEALING	0	0	0	0	1	2012
INTERNATIONAL JOURNAL OF RADIATION ONCOLOGY BIOLOGY PHYSICS	0	0	0	0	1	2015
JOURNAL OF CEREBRAL BLOOD FLOW AND METABOLISM	0	0	0	0	1	2019
JOURNAL OF NEUROLOGICAL SCIENCES-TURKISH	0	0	0	0	1	2012
JOURNAL OF NEUROLOGICAL SURGERY PART B-SKULL BASE	0	0	0	0	1	2020
JOURNAL OF THE NEUROLOGICAL SCIENCES	0	0	0	0	1	2015
MOLECULAR THERAPY	0	0	0	0	1	2018
SEMINARS IN INTERVENTIONAL RADIOLOGY	0	0	0	0	1	1999
STROKE	0	0	0	0	1	2000
SURGICAL LAPAROSCOPY ENDOSCOPY & PERCUTANEOUS TECHNIQUES	0	0	0	0	1	2001

**Table 5 life-12-00175-t005:** The 100 most cited research articles related to Intraoperative MRI.

Paper	DOI	Total Citations	TC per Year
ZOU KH, 2004, ACAD RADIOL	10.1016/S1076-6332(03)00671-8	656	38.59
BLACK PM, 1997, NEUROSURGERY	10.1097/00006123-199710000-00013	577	24.04
SENFT C, 2011, LANCET ONCOL	10.1016/S1470-2045(11)70196-6	403	40.30
NIMSKY C, 2000, NEUROSURGERY	10.1097/00006123-200011000-00008	364	17.33
NABAVI A, 2001, NEUROSURGERY	10.1097/00006123-200104000-00019	328	16.40
CLAUS EB, 2005, CANCER-AM CANCER SOC	10.1002/cncr.20867	312	19.50
NIMSKY C, 2005, NEUROSURGERY	10.1227/01.NEU.0000144842.18771.30	287	17.94
STEINMEIER R, 1998, NEUROSURGERY	10.1097/00006123-199810000-00005	264	11.48
TRONNIER VM, 1997, NEUROSURGERY	10.1097/00006123-199705000-00001	236	9.83
BLACK PM, 1999, NEUROSURGERY	10.1097/00006123-199909000-00001	230	10.45
KUBBEN PL, 2011, LANCET ONCOL	10.1016/S1470-2045(11)70130-9	215	21.50
FERRANT M, 2001, IEEE T MED IMAGING	10.1109/42.974933	208	10.40
SUTHERLAND GR, 1999, J NEUROSURG	10.3171/jns.1999.91.5.0804	204	9.27
HALL WA, 2000, NEUROSURGERY	10.1097/00006123-200003000-00022	191	9.10
NIMSKY C, 2004, RADIOLOGY	10.1148/radiol.2331031352	184	10.82
KUHNT D, 2011, NEURO-ONCOLOGY	10.1093/neuonc/nor133	172	17.20
NIMSKY C, 2006, NEUROIMAGE	10.1016/j.neuroimage.2005.11.001	165	11.00
WIRTZ CR, 2000, NEUROSURGERY	10.1097/00006123-200005000-00017	164	7.81
MAURER CR, 1998, IEEE T MED IMAGING	10.1109/42.736050	160	6.96
CLATZ O, 2005, IEEE T MED IMAGING	10.1109/TMI.2005.856734	158	9.88
NIMSKY C, 2005, RADIOLOGY	10.1148/radiol.2341031984	158	9.88
HADANI M, 2001, NEUROSURGERY	10.1097/00006123-200104000-00021	156	7.80
FAHLBUSCH R, 2001, J NEUROSURG	10.3171/jns.2001.95.3.0381	155	7.75
STARR PA, 2010, J NEUROSURG	10.3171/2009.6.JNS081161	150	13.64
NIMSKY C, 2006, NEUROL RES	10.1179/016164106 × 115125	138	9.20
BOHINSKI RJ, 2001, NEUROSURGERY	10.1097/00006123-200104000-00007	136	6.80
BOHINSKI RJ, 2001, NEUROSURGERY-a	NA	134	6.70
NIMSKY C, 2001, SURG NEUROL	10.1016/S0090-3019(01)00628-0	132	6.60
SCHWARTZ RB, 1999, RADIOLOGY	10.1148/radiology.211.2.r99ma26477	130	5.91
HATIBOGLU MA, 2009, NEUROSURGERY	10.1227/01.NEU.0000345647.58219.07	128	10.67
HALL WA, 1999, NEUROSURGERY	10.1097/00006123-199904000-00067	127	5.77
NIMSKY C, 2006, NEUROSURGERY	10.1227/01.NEU.0000219198.38423.1E	118	7.87
MORIARTY TM, 1996, NEUROSURG CLIN N AM	NA	116	4.64
NIMSKY C, 2001, NEUROSURGERY	10.1097/00006123-200105000-00023	113	5.65
NIMSKY C, 2004, NEUROSURGERY	10.1227/01.NEU.0000129694.64671.91	111	6.53
SCHWARTZ TH, 2006, NEUROSURGERY	10.1227/01.NEU.0000193927.49862.B6	94	6.27
SCHNEIDER JP, 2005, NEURORADIOLOGY	10.1007/s00234-005-1397-1	94	5.88
BARONE DG, 2014, COCHRANE DB SYST REV	10.1002/14651858.CD009685.pub2	92	13.14
KAIBARA T, 2000, NEUROSURGERY	NA	78	3.71
GERLACH R, 2008, NEUROSURGERY	10.1227/01.NEU.0000312362.63693.78	77	5.92
SCHULDER M, 2001, J NEUROSURG	10.3171/jns.2001.94.6.0936	77	3.85
MORIARTY TM, 2000, NEUROSURGERY	10.1097/00006123-200011000-00023	77	3.67
BERNSTEIN M, 2000, NEUROSURGERY	10.1097/00006123-200004000-00023	77	3.67
EYUPOGLU IY, 2013, NAT REV NEUROL	10.1038/nrneurol.2012.279	76	9.50
FAHLBUSCH R, 2005, EUR J ENDOCRINOL	10.1530/eje.1.01970	76	4.75
HALL WA, 2008, J MAGN RESON IMAGING	10.1002/jmri.21273	75	5.77
HOBBS SK, 2003, J MAGN RESON IMAGING	10.1002/jmri.10395	75	4.17
RUBINO GJ, 2000, NEUROSURGERY	10.1097/00006123-200003000-00023	75	3.57
GASSER T, 2005, NEUROIMAGE	10.1016/j.neuroimage.2005.02.022	74	4.62
MARTIN AJ, 2005, MAGN RESON MED	10.1002/mrm.20675	73	4.56
HENSON JW, 2005, LANCET ONCOL	10.1016/S1470-2045(05)01767-5	73	4.56
JOLESZ FA, 1998, JMRI-J MAGN RESON IM-a	10.1002/jmri.1880080104	73	3.17
MARTIN AJ, 2000, RADIOLOGY	10.1148/radiology.215.1.r00ap31221	71	3.38
RODER C, 2014, EJSO-EUR J SURG ONC	10.1016/j.ejso.2013.11.022	70	10.00
ALMEIDA JP, 2015, CURR NEUROL NEUROSCI	10.1007/s11910-014-0517-x	69	11.50
TSUGU A, 2011, WORLD NEUROSURG	10.1016/j.wneu.2011.02.005	69	6.90
MIRZADEH Z, 2014, MOVEMENT DISORD	10.1002/mds.26056	68	9.71
EYUPOGLU IY, 2012, PLOS ONE	10.1371/journal.pone.0044885	68	7.56
SHERMAN JH, 2011, CURR NEUROL NEUROSCI	10.1007/s11910-011-0188-9	68	6.80
NIMSKY C, 2005, ACAD RADIOL	10.1016/j.acra.2005.05.020	68	4.25
WIRTZ CR, 1997, STEREOT FUNCT NEUROS	10.1159/000099900	67	2.79
HERVEY-JUMPER SL, 2014, CURR TREAT OPTION NE	10.1007/s11940-014-0284-7	66	9.43
JOLESZ FA, 2005, NEUROSURG CLIN N AM	10.1016/j.nec.2004.07.011	66	4.12
NIMSKY C, 2002, EUR RADIOL	10.1007/s00330-002-1363-9	66	3.47
TALOS IF, 2006, RADIOLOGY	10.1148/radiol.2392050661	65	4.33
COBURGER J, 2014, NEUROSURG FOCUS	10.3171/2013.11.FOCUS13463	64	9.14
NIMSKY C, 2007, NEUROSURGERY	10.1227/01.NEU.0000144842.18771.30	63	4.50
HIRSCHBERG H, 2005, MINIM INVAS NEUROSUR	10.1055/s-2004-830225	62	3.88
REES J, 2003, CURR OPIN NEUROL	10.1097/00019052-200312000-00001	62	3.44
PRABHU SS, 2011, J NEUROSURG	10.3171/2010.9.JNS10481	60	6.00
BERNAYS RL, 2002, J NEUROSURG	10.3171/jns.2002.97.2.0354	58	3.05
JOLESZ FA, 2001, J MAGN RESON IMAGING	3.0.CO;2-2 “ target = “_blank” > 10.1002/1522-2586(200101)13:13.0.CO;2-2	58	2.90
PAMIR MN, 2010, J NEUROSURG	10.3171/2009.3.JNS081139	57	5.18
HALL WA, 1998, PEDIATR NEUROSURG	10.1159/000028732	56	2.43
COBURGER J, 2016, NEUROSURGERY	10.1227/NEU.0000000000001081	55	11.00
HU JW, 2007, J NEUROSURG	10.3171/jns.2007.106.1.164	55	3.93
KUHNT D, 2011, NEUROSURGERY	10.1227/NEU.0b013e318225ea6b	53	5.30
SENFT C, 2010, CLIN NEUROL NEUROSUR	10.1016/j.clineuro.2009.12.003	53	4.82
KANNER AA, 2002, J NEUROSURG	10.3171/jns.2002.97.5.1115	52	2.74
STARR PA, 2014, J NEUROSURG-PEDIATR	10.3171/2014.6.PEDS13605	51	7.29
LEWIN JS, 1999, AM J NEURORADIOL	NA	51	2.32
BUCHFELDER M, 2002, EPILEPSIA	10.1046/j.1528-1157.2002.46201.x	50	2.63
JOLESZ FA, 1998, JMRI-J MAGN RESON IM	10.1002/jmri.1880080105	50	2.17
ALEXANDER E, 1997, STEREOT FUNCT NEUROS	10.1159/000099896	50	2.08
JENKINSON MD, 2007, BRIT J NEUROSURG	10.1080/02688690701642020	49	3.50
HALL WA, 2006, J NEURO-ONCOL	10.1007/s11060-005-9046-4	49	3.27
MCPHERSON CM, 2004, J NEURO-ONCOL	10.1023/B:NEON.0000024744.16031.e9	49	2.88
WU JS, 2009, NEUROSURGERY	10.1227/01.NEU.0000348549.26832.51	47	3.92
MAMATA Y, 2001, J MAGN RESON IMAGING	3.0.CO;2-X” target = “_blank” > 10.1002/1522-2586(200101)13:13.0.CO;2-X	47	2.35
COBURGER J, 2015, PLOS ONE	10.1371/journal.pone.0131872	46	7.67
KUHNT D, 2012, NEUROSURGERY	10.1227/NEU.0b013e318237a807	46	5.11
FAHLBUSCH R, 2000, CHILD NERV SYST	10.1007/s003810000344	46	2.19
FERRANT M, 2000, LECT NOTES COMPUT SC	NA	46	2.19
HATA N, 2000, J COMPUT ASSIST TOMO	10.1097/00004728-200007000-00004	45	2.14
FUJII M, 2015, J NEUROSURG	10.3171/2014.10.JNS14945	43	7.17
BERKMANN S, 2012, ACTA NEUROCHIR	10.1007/s00701-012-1285-5	42	4.67
SCHULZ T, 2004, EUR RADIOL	10.1007/s00330-004-2496-9	40	2.35
LUDECKE DK, 2006, NEUROENDOCRINOLOGY	10.1159/000095533	0	0.00
AGHI MK, 2015, JOURNAL OF NEURO-ONCOLOGY	10.1007/s11060-015-1867-1	0	0.00
SENFT C, 2008, NEUROSURGERY	10.1227/01.NEU.0000313624.77452.3C	0	0.00

## Data Availability

All data provided in manuscript.

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
