# Peer review of "Evaluating the Impact of Intraoperative MRI in Neuro-Oncology by Scientometric Analysis"

_life, 2022, doi:10.3390/life12020175_

Round 1

Reviewer 1 Report

The authors conducted a detailed Bibliometric analysis of the academic impact and publication status of Intraoperative Magnetic Resonance Imaging. Below are my comments.

1) Since it is a computer-based comprehensive search, Bibliometric analysis visualizes the trends that are not visible with “human-hand search”. It can be further emphasized that bibliometric analysis may sometimes be superior to or more detailed than “human-hand search”. This also justifies the authors' choice of this method for their research. It can be mentioned somewhere.

2) Although authors wrote that “There have been a few bibliometric analyses conducted previously in Neurosurgery”, the MDPI also published a detailed original article on bibliometric analysis in Neurosurgery from Cancers in 2019.

https://www.mdpi.com/2072-6694/11/2/178

This can be cited as a study of one Bibliometric analysis in neurosurgery.

3) I am not sure the method how to analyze the “Most relevant words” in Supplementary figure 1 was described in this manuscript. Had any text mining method been used? This point should be written.

4) Authors wrote as below. “Interestingly a shift of trends from “Image guided surgery’ and ‘accuracy’ in the early 2000s to ‘extent of resection’, ‘impact’ and ‘survival’ in the later years, was noted.” Since this was written in abstract, this part should be very important or “neues” part of this study. However, I couldn’t find a detailed description of it in the result section. This may also be related to my third point above. I could find some fragmentary descriptions in line 183 or line 269, but they were not so detailed. If authors thought “interestingly,” and want to emphasize this finding, it should be focused in the result section, and also the figure should be in the normal figure, not a supplementary figure. Restructuring of the description of these findings can be considered.

5) The authors analyzed the top 100 most cited papers. On the other hand, it is known that the number of citations for each paper will increase over the years. The number of citations of the latest papers is small. The “Average years from publication” raw in Table 1 also suggests us this point. This means that the analysis of the top 100 most cited papers may exclude the latest-trend papers unintentionally and selectively. This method may cause the obsolescence of this study's analysis. This point can be a limitation of this study and should be mentioned somewhere.

6) Readers of this article may think that the research of Intraoperative Magnetic Resonance Imaging became no longer been the mainstream trend since 2017, seeing Figure 1A. An understandable counterargument to this view can be stated somewhere.

Reviewer 2 Report

Dear authors,

You conducted a first bibliomatric analysis on IOMRI, and you describe a shift from the focus of accuracy in the early years towards a focus on extent of resection and impact  and survival in more recent years. Your methodology is well illustrated. Interesting remarks are made on the blind spots in the literature of the pediatric population and the posterior fossa. This could stimulate pediatric neurosurgeons since the extent of resection of pediatric brain tumors still affects prognosis both in survival and quality of life.

Your paper contains interesting information for the clinicians in this field.   

Author Response

Dear authors,

You conducted a first bibliometric analysis on IOMRI, and you describe a shift from the focus of accuracy in the early years towards a focus on the extent of resection and impact and survival in more recent years. Your methodology is well illustrated. Interesting remarks are made on the blind spots in the literature of the pediatric population and the posterior fossa. This could stimulate pediatric neurosurgeons since the extent of resection of pediatric brain tumors still affects prognosis both in survival and quality of life.

Your paper contains interesting information for clinicians in this field. 

REPLY

We thank you for the magnanimous review and hope to stimulate more research in this field, especially in pediatrics. 

Reviewer 3 Report

The paper is well written and interesting. It offers a new point of view in the field of research. 

Author Response

The paper is well-written and interesting. It offers a new point of view in the field of research. 

REPLY

We thank you for the review and hope that our contribution will be impactful. 

Reviewer 4 Report

Deora et al. performed a scientometric analysis to evaluate the impact of Intraoperative Magnetic Resonance Imaging (IOMRI) in neuro-oncology. This article gives a whole new perspective of Bibliometric analysis to elucidate the importance of IOMRI in neuro-oncology. However, some questions/issues should be answered.

1.The abstract paragraph format is different from the rest parts of the manuscript.

2.In line 65 methods part, “Search Strategy” is not aligned with the rest of the titles.

3.In the abstract, authors described that they did a title-specific search on 4th November 2020, but in the method-search strategy, authors described that they did a title-specific search on 23rd April 2021 (line67-68).

4.Line 40 needs the reference for “The development of the first intraoperative MRI, in 1991”.

5.The authors described that the development of the first intraoperative MRI was in 1991. In figure 1A, the authors showed the graph demonstrating “annual production of ALL articles on intraoperative MRI”. Based on figure 1A, there is an article on intraoperative MRI published in 1987. Moreover, in figure 1C, there seem to be 0.5 citations in 1987. The authors need to explain this 1987 article.

6.There is no list in supplemental data about the 663 articles that met the inclusion criteria and were included in the final analysis in the manuscript. There is no list of what the 100 most cited articles are. Furthermore, authors should provide whether these analyzed articles are randomized control trials or not.

7.What is the exclusion criteria of this analysis? It is unclear based on the current manuscript now.

8.What are the impact factors of the journals analyzed in this manuscript?

9.The format of the figures is different from the Life journal. For example, in figure 1A, it is weird to read the title like “Annual Scientific Production 1A” (line 122)—same issues for all the rest of the figures.

10.In the conclusion part, the authors claimed that this bibliometric analysis “highlighting ‘blind spots’ in research such as the exclusion of paediatric populations, spinal tumours and posterior fossa fossa pathologies”(I assume “posterior fossa fossa pathologies” means “posterior fossa pathologies” ). However, in this manuscript, the authors talked very little about pediatric populations, spinal tumors and posterior fossa pathologies in the manuscript, and there are no analyses from this manuscript on these topics.

Round 2

Reviewer 4 Report

Based on the authors’ response to the comments:
Questions 1-3 are not corrected at all.
Question 4, after adding the new reference, there are two references #1 in the reference list.
Question 6, where is “table 5”? Please list the 663 articles in the supplemental data.
Question 8, table 2 does not include impact factors.
Question 10, the authors still use “posterior fossa fossa pathologies” in conclusion, after it is pointed out. Can authors explain what is “posterior fossa fossa pathologies”?

Author Response

Questions 1-3 are not corrected at all.

REPLY: Abstract is in line with the manuscript. The date everywhere is 23rd April. Line alignment has been corrected. 

Question 4, after adding the new reference, there are two references #1 in the reference list.

Reply: There is only one #1 reference: "Seifert V. Intraoperative MRI in neurosurgery: technical overkill or the future of brain surgery? Neurol India. 2003 Sep;51(3):329-32. PMID: 14652431." 

Question 6, where is “table 5”? Please list the 663 articles in the supplemental data.

663 ARTICLES ARE LISTED IN SUPPLEMENTARY TABLE 3 NOW. Table 5 had been submitted previously and has not been revised. 

Question 8, table 2 does not include impact factors.

Table 2 has all the names of the journals and impact factors are not listed as they change constantly and hence can be looked up whenever required. 

Question 10, the authors still use “posterior fossa fossa pathologies” in conclusion, after it is pointed out. Can authors explain what is “posterior fossa fossa pathologies”?

REply: "Posterior fossa pathologies" included pathologies reported in the posterior fossa i.e medulloblastoma, ependymoma, vestibular schwannoma, cerebellar gliomas. Any tumor which needs a prone position has been difficult to execute with intraoperative MRI. That's why the use of this term. 

Its available in medical encyclopedia as a usable term: "https://medlineplus.gov/ency/article/001404.htm"